# Current and Future Horizons of Patient-Derived Xenograft Models in Colorectal Cancer Translational Research

**DOI:** 10.3390/cancers11091321

**Published:** 2019-09-06

**Authors:** Akira Inoue, Angela K. Deem, Scott Kopetz, Timothy P. Heffernan, Giulio F. Draetta, Alessandro Carugo

**Affiliations:** 1Department of Genomic Medicine, UT MD Anderson Cancer Center, Houston, TX 77054, USA; 2Department of Gastroenterological Surgery, Graduate School of Medicine, Osaka University, Osaka 565-0871, Japan; 3Institute for Applied Cancer Science, UT MD Anderson Cancer Center, Houston, TX 77054, USA; 4Department of Gastrointestinal Medical Oncology, UT MD Anderson Cancer Center, Houston, TX 77030, USA; 5Moon Shots ProgramTM, UT MD Anderson Cancer Center, Houston, TX 77030, USA; 6TRACTION Platform, UT MD Anderson Cancer Center, Houston, TX 77054, USA

**Keywords:** precision medicine, drug development, avatar models, co-clinical trials, humanized mice

## Abstract

Our poor understanding of the intricate biology of cancer and the limited availability of preclinical models that faithfully recapitulate the complexity of tumors are primary contributors to the high failure rate of novel therapeutics in oncology clinical studies. To address this need, patient-derived xenograft (PDX) platforms have been widely deployed and have reached a point of development where we can critically review their utility to model and interrogate relevant clinical scenarios, including tumor heterogeneity and clonal evolution, contributions of the tumor microenvironment, identification of novel drugs and biomarkers, and mechanisms of drug resistance. Colorectal cancer (CRC) constitutes a unique case to illustrate clinical perspectives revealed by PDX studies, as they overcome limitations intrinsic to conventional ex vivo models. Furthermore, the success of molecularly annotated "Avatar" models for co-clinical trials in other diseases suggests that this approach may provide an additional opportunity to improve clinical decisions, including opportunities for precision targeted therapeutics, for patients with CRC in real time. Although critical weaknesses have been identified with regard to the ability of PDX models to predict clinical outcomes, for now, they are certainly the model of choice for preclinical studies in CRC. Ongoing multi-institutional efforts to develop and share large-scale, well-annotated PDX resources aim to maximize their translational potential. This review comprehensively surveys the current status of PDX models in translational CRC research and discusses the opportunities and considerations for future PDX development.

## 1. Introduction

Oncology drug discovery suffers the highest percentage of clinical trial failures among all disease types [1]. Thirty-two percent of combined phase II and phase III clinical trials between 2013 and 2015 failed, up from 23% in 2010, underscoring the inadequacy of preclinical models to predict drug response in the clinical setting. Recent, systematic investigations of therapeutic targets or dependencies across a large number of diverse, annotated human cancer cell lines have confirmed significant liabilities with regard to the clinical translation of data derived using these models [2,3,4], including significant genetic, epigenetic, and transcriptomic changes resulting from the adaptation of these cells to grow in artificial culture conditions [5,6]. Further, these cell lines no longer maintain the complex heterogeneity of the primary tumor, having lost, or been enriched for specific subclones, in addition to missing relevant components of the human tumor microenvironment [7]. As a result, the US National Cancer Institute (NCI) has deprioritized the NCI-60 human cancer cell line panel for most drug screening [8]. Increasingly, patient-derived xenograft (PDX) models generated by direct engraftment of human tumor tissues obtained through biopsy or surgical resection into immunodeficient rodents are the models of choice to address clinically relevant questions in cancer research [9,10,11,12,13,14]. With the diverse array of well-characterized PDX models now in use, it is apparent that tumors established from PDXs recapitulate in situ primary human tumors more accurately compared to cell line models with regard to tumor architecture, microenvironment, and heterogeneity [15,16]. Even after several passages, PDX models maintain key molecular, genetic, and histological characteristics of the original human tumor from which they were derived [17,18,19]. These are powerful tools for understanding the biology of human tumors, providing translational opportunities for analysis of predictive biomarkers, therapeutic targets, and drug discovery for precision medicine in cancer [20,21].

CRC is a significantly heterogeneous disease with distinct variations in molecular features and responses to therapy [14,22,23,24,25,26,27,28,29,30,31,32,33]. Preclinical models that faithfully recapitulate the complexity of human tumors are needed to successfully translate our findings from bench to bedside. We have witnessed major advancements in the sophistication and application of PDX models to improve our understanding of tumor biology and mechanisms of drug response in CRC [34,35,36,37,38,39,40]. Although PDX models are considered to be complementary to other preclinical models, such as genetically engineered mice (GEMs) that are appropriate to address specific questions within a well-defined genetic background, PDX models in CRC are well suited to address numerous clinical scenarios. CRC PDX models have been successfully applied to study tumor heterogeneity, clonal dynamics, and the tumor microenvironment, and they have aided substantially in drug discovery by improving the translatability of co-clinical trials and biomarkers. However, treatment options for patients with CRC remain limited, and there is substantial room to improve clinical outcomes. Our ability to develop cancer models that better replicate the complex biology of human tumors is essential to continue to push the field beyond a traditional reductionist approach to facilitate successful translation of scientific findings into clinical practice. In this review, we will assess how current resources and trends in using PDX models are already advancing the field of CRC translational research and look toward opportunities to further develop and optimize these models to hasten our delivery of novel clinical approaches to manage CRC, and cancer treatment at large (Figure 1).

## 2. PDXs as Bona Fide Genetic and Histologic Mimics of CRC

The establishment of fully annotated PDX models that recapitulate the complex human tumor has become essential for translational cancer research. To this end, extensive multi-institutional efforts are now underway to establish and characterize a large collection of PDX models at the molecular and histopathologic levels to ensure that they represent the diversity of clinical cases [10]. In CRC, histopathologic characterization of PDXs has confirmed a high concordance between PDXs and the corresponding primary tumor in terms of tumor differentiation, mucus secretion, and stromal composition, as well as maintenance of primary intratumoral clonal heterogeneity [14,22,23,24,25,35,38,41]. One study showed that CRC PDXs retain the chromosomal instability of early stage CRC tumors up to 14 passages [24]. Another study using standard clinical parameters demonstrated that CRC PDXs were histopathologically classified as well to moderately differentiated (82%), moderately differentiated to undifferentiated (8%), or mucinous adenocarcinomas (10%) [22]. It has been shown that tumor cells are clearly enriched in xenografts compared to patient samples, and this enrichment was observed after the first passage; but, once the xenograft is established, the proportion of stroma in the tumor remained stable among the different passages [22]. Notably, an in situ hybridization technique using human-specific probes showed the complete loss of human stroma from the xenograft in early passages [22]. Overall, these observations corroborate that CRC PDXs faithfully recapitulate the histopathologic features and diversity of human CRC tumors up to a certain passage.

Comprehensive genetic characterization of PDXs in CRC showed that the frequency of common genetic mutations in PDXs is similar to that reported in primary tumors [14,22,23,25,38,41,42,43,44]. Yao et al. [45], in a molecular characterization of 79 CRC PDX models by whole-exome and RNA sequencing as well as SNP6.0 analysis, showed that key genetic mutations and their frequencies, such as *APC* (75.9%), *p53* (70.9%), *KRAS* (55.7%), *NRAS* (5.1%), *BRAF* (15.2%), *PIK3CA* (26.6%), *PTEN* (12.7%), *PIK3R1* (6.3%), and *CTNNBI* (3.8%), were consistent with the data reported for human CRC tumors [45,46]. In addition, Bertotti et al. [14] established CRC PDXs from 85 metastatic CRC samples and showed that 2 of the 85 (2.3%) displayed high-grade microsatellite instability (MSI), consistent with the 1.3% to 2.7% frequency reported in metastatic CRC [47,48]. This study also detected largely concordant copy number variations between first- and second-passage PDXs and their matched original counterparts, but a marked heterogeneity among tumors from different patients. Another study investigated epigenetic dynamics in PDXs in osteosarcoma and CRC, and revealed that an average 2.7% of the assayed CpG sites undergo major (Δβ ≥ 0.51) methylation changes in a cancer-specific manner as a result of the xenografting procedure, without any significant subsequent methylation changes between primary and secondary PDXs [49]. Overall, these findings confirm that the genetic and epigenetic features of primary patient tumors are maintained through passages in PDX models of CRC. 

## 3. PDXs Recapitulate Molecularly Defined CRC Subtypes

Technological advancements allowing extensive characterization of CRC PDXs based on the proteome, metabolome, and microbiome as well as the immune microenvironment have provided a unique opportunity to improve patient care in the context of precision medicine [34,36,37]. Critical driver genomic events in CRC have been extensively investigated, and several genes and pathways have been shown to be involved in CRC tumor initiation and growth. However, recent advances in our understanding of genomic and transcriptomic analyses of CRC have uncovered an even more complex and heterogeneous cancer genome than previously appreciated, in addition to complex clonal evolution patterns that emerge under selective pressure. 

A new classification in CRC developed by Isella et al. [26] examined a large collection (*n* = 515 samples from 244 patients) of PDXs in which stromal components of the original tumors were substituted by their murine counterparts, so that detection of cancer cell-specific transcriptomes was possible using human-specific arrays. Through this approach, five CRC intrinsic subtypes (CRIS) were identified: (i) CRIS-A: mucinous, glycolytic, enriched for MSI or *KRAS* mutations; (ii) CRIS-B: transforming growth factor β (TGF-β) pathway activity, epithelial-mesenchymal transition (EMT), poor prognosis; (iii) CRIS-C: elevated EGFR signaling, sensitivity to EGFR inhibitors; (iv) CRIS-D: WNT activation, *IGF2* gene overexpression and amplification; and (v) CRIS-E: Paneth cell-like phenotype, *TP53* mutations. CRIS subtypes successfully stratify independent sets of primary and metastatic CRCs by minimizing the confounding effects of stromal-derived intratumoral heterogeneity [50], with limited overlap on existing transcriptional classes and unprecedented prognostic performance. These studies have provided critical insights into our understanding of the cancer cell-intrinsic and stromal components of CRC. 

In another effort to classify CRC, the International Colorectal Cancer Subtyping Consortium identified four robust consensus molecular subtypes (CMS) of CRC from a large-scale analysis of gene expression data from The Cancer Genome Atlas [27,28,29,30,31,32,33,51] that more comprehensively describe CRC heterogeneity at the global gene-expression level versus other approaches. For instance, *RAS* wild-type tumors, which were previously considered to be a homogenous population, are found across distinct CMS groups with profound biological differences, leading to heterogeneous drug responses. Thus, CMS enable more precise positioning of targeted therapies by considering biological behavior in addition to genotype: (i) CMS1 (14%) is characterized by a diffuse immune infiltrate with features of MSI; (ii) CMS2 (37%) displays epithelial differentiation and strong upregulation of WNT and MYC targets; (iii) CMS3 (13%) displays marked metabolic deregulation with overrepresentation of *KRAS* mutations; and (iv) CMS4 (23%) displays mesenchymal features, TGF-β activation, and extensive stromal invasions, and is associated with poor prognosis. 

Establishing a large cohort of CRC PDX models representing each molecular subtype could be a powerful tool to identify subtype-specific therapeutic targets and biomarkers to predict response to therapy [52]. For instance, Medico et al., in a report of genetic and transcriptional profiling of 151 CRC cell lines, demonstrated that CRC transcriptional subtypes are maintained in these cell lines, and that outlier kinase genes are clinically actionable targets in CRC [53]. A similar approach could be applied to CRC PDX models, using genetic and transcriptional profiling to identify subtype-specific actionable targets. Such studies focused on CMS4 mesenchymal tumors are of particular interest, as this subtype has the worst prognosis and a high risk of metastasis [51]. CMS4 tumors are characterized by activation of pathways related to EMT and stemness, such as TGF-β and integrins, and are mostly derived from prominent stromal cell infiltration of adjacent cancer tissue, particularly cancer-associated fibroblasts (CAFs) [51,54,55]. Recent studies indicate that high expression of the stromal signatures plays an important role in mediating resistance of cancer cells to chemotherapies and targeted agents in CRC [55,56]. Indeed, retrospective analysis of a randomized clinical study showed that the CMS4 subtype was associated with a lack of benefit from adjuvant oxaliplatin-based chemotherapy [57]. It has been shown that the use of TGF-β signaling inhibitors in CRC PDX models to block the crosstalk between cancer cells and the microenvironment reduced the number of metastases, demonstrating a dependency on TGF-β signaling in stromal cells during metastasis in CRC [54]. These findings suggest important insights into the critical contributions of stromal components in CRC development and management.

How and to what extent cancer cell-intrinsic gene expression traits contribute to CRC subtype remain elusive. Because stromal components present in whole-tumor transcriptomes are a dominant source of variation that can mask the biologically relevant transcriptional features of cancer cells, CRC PDX models generated from stromal-dominant tumors, such as ‘Stem/Serrated/Mesenchymal’ (SSM) transcriptional tumors or CMS4 mesenchymal tumors, may not maintain their features or subtype, because human stromal components are replaced by murine counterparts after the initial engraftment. To address this, Sveen et al. developed an optimized classifier enriched for cancer cell-intrinsic gene expression signals and performed high-throughput drug screening to identify subtype-specific sensitivity in CRC [58]. They demonstrated that the cancer cell-adapted classifier performed well in primary CRC tumors, and 148 CRC cell lines and 32 CRC PDX models were shown to recapitulate the biology of the CMS groups. They also demonstrated that CMS2 tumors have strong responses to EGFR and HER2 inhibitors, while CMS1 and CMS4 groups have strong sensitivity to HSP90 inhibitors. These findings and others suggest translational opportunities for CMS classification in preclinical models for identification of subtype-specific drugs in CRC [59].

## 4. Diversity of PDX Models Needed to Test Translational Hypotheses 

The management of a large collection of PDXs entails extensive labor and resources; thus, careful technical and biological considerations should be given to the model chosen to ensure its features adequately address the experimental question. It typically takes 1 to 4 months to establish a PDX model and 4 to 8 months to expand a sufficient number of mice to perform an interpretable preclinical study [60]. 

### 4.1. Heterotopic vs. Orthotopic Models in CRC

In the establishment of PDX models, freshly resected primary human tumor is implanted into the host directly, either as small pieces or as a tumor cell suspension, alone or with matrigel. For CRC PDXs, the most common site of implantation is subcutaneously into the host’s back or flank, because this site facilitates engraftment, monitoring, and resecting the tumor. Some groups have successfully established orthotopic CRC PDX models by implanting tumor cells into the cecum wall or kidney subcapsules of mice; these models offer the advantage of maintaining the microenvironment for the study of the metastatic capacity of the tumor cells [22,38,39]. Rashidi et al. developed a highly metastatic orthotopic transplant nude mouse model of a liver metastasis from a human CRC patient, in which all mice implanted with the tumor had liver metastasis within 10 days, and lymph nodes draining the liver showed metastasis 19 days after implantation [39]. Fujii et al. established matched pairs of primary and metastatic organoids from CRC patients and xenotransplanted them into the kidney subcapsules of immunodeficient (NOD/Shi-*scid*/IL2Rg^null^; NOG) mice. Interestingly, metastasis-derived CRC organoids from this model exhibited higher metastatic capacity than matched primary tumor-derived organoids [41].

Orthotopic models offer a robust platform to study the biology of metastasis and response to therapy in CRC. However, because orthotopic implantation into the colorectal wall of the host requires considerable technical skill, this model offers limited capacity and reproducibility. On the other hand, subcutaneous models, once established, can be conveniently characterized, used for primary cell line or organoid generation, passaged for expansion, or cryopreserved for future study. With recent technological advancements, circulating tumor cells (CTCs) or fluid from malignant ascites or pleural effusions have also been used to generate PDXs that successfully maintain tumorigenicity in mice [21,61,62,63,64]. It has been shown that ex vivo cultured CTCs also maintain their tumorigenic potential in CRC [40]. Therefore, CTC-derived PDX models hold promise for the study of the genetic evolution of tumors and evaluation of their responses to novel drugs.

### 4.2. PDX Tumor Engraftment in CRC

The establishment of PDX models is challenging due to a high rate of engraftment failure. It is difficult to predict successful tumor establishment, as it is influenced by many factors, including the particular characteristics of tumor subtypes (see above discussion), host strain, site of implantation (subcutaneous versus orthotopic), viability of tumor cells, number of tumor-initiating cells, primary versus metastatic tumor, patient’s treatment status at the time of resection, sterility of the tumor specimens, time interval from removal of the tumor to implantation, and adherence to proper surgical technique. Previous studies showed that the rate of successful PDX engraftment is relatively high for CRC, ranging from 56% to 87.5% in various studies [18,22,38,65,66,67], compared to other tumors, such as epithelial ovarian cancer (48.8%) [68], breast cancer (27%) [69], and melanoma (28%). For instance, Lee et al. found that brain metastases from non-small cell lung cancer (NSCLC) showed a significantly higher rate of successful PDX establishment than primary specimens (74% vs. 23%) [70]. In CRC, Katsiampoura et al. demonstrated that the success rate of CRC PDX engraftment was higher in surgical specimens (36/50; 72%) versus in biopsy specimens (14/40; 35%) [65]. An additional example, Oh et al. examined PDXs from 241 CRC patients and found that successful engraftment was associated with poor disease-free survival rate in patients with stage III disease (*p* = 0.034) [66]. A systematic review of CRC PDXs indicated that mouse model selection is also critical, with more severely immunosuppressed strains such as NOD/SCID or NOD/SCID/IL2g^null^ (NSG) being more conducive to PDX generation, owing to their lower immune rejection and higher engraftment rates [18,71].

### 4.3. Intratumoral Heterogeneity and Clonal Dynamics

One critical consideration is that PDXs likely do not fully represent the heterogeneity of the primary tumor. Human tumors are comprised of multiple subclones that harbor distinct genetic and epigenetic alterations, making it at least unlikely, if not impossible, that PDXs generated from a small chunk of the original tumor would fully recapitulate the clonal composition of the primary lesion [72,73,74]. In CRC, it has been shown that pathology sampling region and degree of stromal infiltration significantly affect the molecular classification [75]. Similarly, PDXs are generated from a sample taken from a discrete moment in the development of the tumor and, thus, represent only a “snapshot” of the evolving disease. 

Engraftment of tumor cells into a host exerts a dynamic selective pressure that significantly changes the clonal architecture of the original tumor depending on the host microenvironment. It has been documented that clonal and genetic variability can be exacerbated by serial passages of PDXs, producing significantly variable PDXs and divergent responses to any one treatment [74,76]. Eirew et al. recently used a single-cell sequencing technique to demonstrate that the spectrum of initial clonal selection pressure is quite variable, and only a minority of clones propagate in mice to contribute to the xenograft [77]. It has been reported that the majority of these minor clones are undifferentiated tumor-initiating cells with more aggressive cell-autonomous phenotypes, which may be responsible for long-term tumorigenicity [42,78,79]. A comprehensive study that monitored the dynamics of copy number alterations (CNAs) in 1110 PDX samples across 24 cancer types showed rapid accumulation of CNAs during PDX passaging, often due to selection of preexisting minor clones [80]. Importantly, however, the particular CNAs acquired during PDX passaging differed from those selected during tumor evolution in patients. Therefore, these findings should be considered with caution when PDX models are applied for preclinical cancer research.

Critically, studies have demonstrated that variable clonal repopulation dynamics and the genomic stability of PDXs influence chemotherapy response in CRC [80,81,82]. Therefore, to understand the response of tumors to therapeutic intervention, we must not simply measure the tumor size, but also monitor the clonal dynamics of PDXs throughout the experiment. Some of these limitations can be addressed by minimizing the number of passages and optimizing experimental design with statistically adequate power and proper randomization [83]. Another approach has been recently developed by Seth et al., wherein Clonal Replica Tumors (CRTs) were generated in pancreatic PDX models [84]. CRTs enable the generation of large cohorts of mice bearing clonally identical tumors, as well as the isolation and expansion of treatment-naïve subclones to explore the molecular mechanisms underlying variable drug responses. Adaptation of this approach to CRC and other tumor types would directly address many of the concerns related to clonal dynamics and heterogeneity in PDX models. 

### 4.4. Tumor Microenvironment

After engraftment, the human microenvironment is rapidly replaced in a time- and size- dependent manner by murine stromal cells, including CAFs, inflammatory cells, extracellular matrix, and blood/lymphatic vessels [85,86]. This presents a challenge when we try to understand stromal contributions to cancer biology and investigate stroma-mediated resistance to therapeutic strategies that target this compartment [87]. The predominant strategy for overcoming this challenge is the “humanized PDX model.” Humanizing the murine immune system consists of ablating the endogenous immune system followed by transplanting human peripheral blood cells, hematopoietic stem cells, or human tumor-infiltrating lymphocytes into immunodeficient mice [88,89,90]. Although there are species-specific differences in both the adaptive and innate immune systems, this emerging platform could potentially overcome the challenges implicit in immunodeficiency. 

With an increasing number of cancer immunotherapies and vaccines emerging for cancer treatment, humanized PDX models that faithfully evaluate the preclinical activity of these therapies in a context of a functional human immune system are attractive. For instance, immunotherapies such as adoptive transfer of chimeric antigen receptor T cells or autologous tumor-infiltrating T cells [91,92]; antibodies targeting immune checkpoint mediators, including programmed cell death 1 (PD1), PD ligand 1 (PD-L1), or cytotoxic T lymphocyte–associated antigen 4 (CTLA-4) [93,94,95]; and cancer-specific peptide vaccine-based immunotherapies [96,97,98] have shown promising antitumor activities in humanized or preclinical cancer models. In addition, recent studies have demonstrated that high-throughput screening approaches using genome-scale shRNAs or CRISPR-Cas9 libraries are useful in identifying immunotherapy targets in human cell lines in vitro or mouse tumors in vivo [99,100,101]. 

Another important application of humanized PDX models is examination of interactions between the immune system and the microbiome. Recent evidence suggests that the human gut microbiome greatly contributes to the progression of CRC through the procarcinogenic activities of specific pathogens and their metabolites [37,102,103,104]. A study in melanoma showed that a commensal strain of Bifidobacterium promotes antitumor immunity and facilitates anti-PD-L1 efficacy, suggesting that manipulation of the microbiome may modulate the action of cancer immunotherapy [105]. It is also possible to alleviate cancer drug toxicity by inhibiting a bacterial enzyme [106]. Therefore, reconstitution of both a functional human immune system and microbiome in PDX models could provide an opportunity to better understand the complex biology of the immune system, the microbiome, and metabolites.

## 5. Application of PDX Models in CRC Translational Research

### 5.1. Drug Discovery

Although the development of cancer cell-line culture techniques enabled cancer biology discovery, our excitement to use these models to translate biological findings to clinical practice has demonstrated their limitations, which has in turn fueled interest in PDX models with greater predictive value for incipient drug testing [6,9]. Cell lines derive from cancer cells that have adapted to growth in in vitro culture, resulting in selection of adaptations distinct from the genetic stress imposed on tumors in situ or in vivo [107]. Furthermore, there is generally minimal stromal infiltration into the homogeneous cell masses that form from cell line-based xenografts, which makes these relatively poor models of the primary tumor architecture and inter-cellular interactions that are critical in promoting tumor progression [108]. In CRC, a pronounced desmoplastic reaction serves as an independent indicator of poor prognosis and cancer recurrence [109]. It is also increasingly appreciated that, in solid tumors such as pancreatic cancer, the desmoplastic reaction decreases intratumoral drug perfusion, which may alter the pharmacokinetics of therapeutic agents [110] and result in overestimation of the antitumor effects. So, the prime rationale for leveraging PDX models in oncology research is these models’ faithful recapitulation of human cancer biology and prediction of clinical outcomes in patients. Accumulating evidence suggests high concordance between outcomes in PDX models and patients with regard to drug response, supporting the use of clinically relevant PDX models to investigate novel drugs or therapeutic targets in CRC [10,22,43,45,111,112,113,114,115,116,117,118,119].

*RAS* mutational status in CRC, for example, has been well documented and is the only clinically approved biomarker for EGFR-targeted therapy. Nunes et al. [43] examined 52 CRC PDXs and showed that responses of these models to cetuximab parallel clinical data in patients, with partial or complete response achieved in 15% (8 of 52) of PDXs, and the response strictly restricted to *KRAS* wild-type models [120]. Julien et al. [22] demonstrated that, in a 54 CRC PDX cohort, mice with wild-type *KRAS* survived longer after treatment with cetuximab than mice with a mutant *KRAS* tumor. Yao et al. [45] evaluated the antitumor efficacy of cetuximab in combination with a pan-RAF inhibitor in 79 CRC PDX models. In this PDX study, concurrent EGFR and RAF inhibition demonstrated synergistic antitumor activity for CRC PDX models with a *KRAS* or *BRAF* mutation, providing a potential treatment option in a clinical trial setting for CRC patients with *KRAS*- or *BRAF*-mutant tumors. Another study showed that combined inhibition of MEK and CDK4/6 was effective in *KRAS*-mutant CRC PDXs and justified a phase II clinical trial in patients with refractory *KRAS*-mutant CRC [25,121]. Moreover, PDX models that responded to single-agent cetuximab had significantly higher *EGFR* copy number and higher EGFR expression, which is again similar to the findings observed in patients [120,122,123]. In contrast, dual inhibition of MEK and PI3K/mTOR showed limited preclinical activity in a cohort of 40 *RAS*-mutant CRC PDXs, which is consistent with data from a phase I clinical trial [124,125,126]. We conclude that preclinical data from PDX models are positively correlated with clinical outcomes in CRC.

### 5.2. Biomarker Identification

Another utility of PDX models to discover biomarkers that may help increase understand of the resistance mechanisms of drugs to advance the design of resistance-overcoming combination therapies. Bertotti et al. [14] established CRC PDX cohorts from 85 patient-derived, genetically characterized metastatic CRC samples and performed prospective trials to identify novel biomarkers of resistance to cetuximab. They found that *HER2* amplification was enriched specifically in a subset of cetuximab-resistant, *KRAS/NRAS/BRAF/PIK3CA* wild-type PDX models and in clinically nonresponsive *KRAS* wild-type tumors. In addition, it has been shown that treatment with HER2-targeted trastuzumab plus tyrosine kinase inhibitors produced regression of *HER2*-mutated CRC PDXs [91]. Indeed, this preclinical study provided a strong rationale for a successful clinical trial targeting *HER2*-activating mutations in patients with CRC [127,128]. *IGF2* overexpression or *MET* amplification is also associated with primary resistance to cetuximab in CRC PDX models, which could potentially be overcome by a combination of IGF2 and MET inhibitors [129,130,131,132]. Comprehensive genomic analyses of tumors from both PDX models and patients whose tumor responded to anti-EGFR antibody blockade detected mutations in *ERBB2, EGFR, FGFR1, PDGFRA*, and *MAP2K1* as potential mechanisms of primary resistance to this therapy [15]. Therapeutic resistance to EGFR blockade could be overcome through combinatorial therapies targeting actionable genes in CRC PDXs. Similarly, the activation of compensatory feedback loops of ERBB3 has been identified as a mechanism of resistance to MEK inhibition in *KRAS*-mutant CRC, which potentially could be overcome by combinatorial treatment [133]. These findings indicate that PDX models play an important role in identifying potential biomarkers of drug efficacy and can facilitate optimization of clinical trial designs.

### 5.3. High-Throughput Pharmacological Screening 

Considering heterogeneity of drug responses, the drug discovery and biomarker studies just described have limited value in predicting clinical trial responses at the population level due to, at least in part, the limited number of PDX models. To overcome this, Gao et al. [134] established a large collection of PDXs comprising 1075 models across the spectrum of common solid cancers that are fully characterized for their mutations, CNAs, and mRNA expression levels. The authors performed a large-scale in vivo drug screening to understand interpatient heterogeneity in responses to 62 treatments with a “one animal per model per treatment” (1 × 1 × 1) preclinical trial setting. The reproducibility and clinical translatability of this approach was demonstrated by identifying associations between a genotype and drug response and established mechanisms of resistance. Moreover, they demonstrated a high degree of consistency between the responses to targeted therapies observed in PDXs and clinical outcomes in cancer patients. The authors also presented an example in which a disconnection between in vitro cell-based assays and in vivo PDX trials was observed. Specifically, LFW527, an IGF1R inhibitor, was identified in a large-scale combination screen as having strong synergy with binimetinib, a MEK inhibitor, in 18 of 45 CRC cell lines. Contrary to what was observed in vitro, the combination of LFW527 with binimetinib did not improve response rate compared to single-agent treatment across 35 CRC PDXs. Although this experimental approach is extremely resource intensive, PDXs could serve as powerful tools for large-scale genotype-phenotype correlation in genetically defined populations. Therefore, this approach could potentially improve preclinical evaluation of treatment modalities and enhance our ability to predict clinical trial responses. 

Recent advances in genome editing technologies using shRNA or CRISPR/Cas9 also enable the identification of novel therapeutic targets in PDX models. Carugo et al. developed an unbiased, in vivo loss-of-function genomic screening platform to identify oncogenic drivers necessary to sustain tumors in pancreatic cancer PDX models [135]. Focusing on epigenetic regulators in this study, the authors identified and validated WDR5, a core member of the COMPASS histone H3 Lys4 (H3K4) MLL (1–4) methyltransferase complex, as an essential gene for pancreatic tumor maintenance in PDX models. Such genetic screening approaches have been widely adopted to identify novel vulnerabilities and therapeutic targets across different PDX models, as well as in organoids and ex vivo systems [99,136,137,138,139,140].

## 6. Is the “Avatar” Co-Clinical Trial an Option in CRC?

To address the failure rate of clinical trials in oncology, preclinical models that accurately predict clinical outcomes are urgently needed. Accumulating evidence shows that PDX models faithfully reflect clinical outcome with regard to drug response and prognosis [14,66,141]. From this body of evidence has emerged the so-called “Avatar” concept for co-clinical trials. PDX Avatar models are generated from the tumors of patients enrolled in a clinical trial, and these models are treated simultaneously with the same agents administered to the patients in the clinical trial [8,142]. Coupled with tumor genomic profiling data, Avatar co-clinical trials are designed to aid in the design of personalized therapeutic regimens in real time. For instance, Corcoran et al. conducted a clinical trial to evaluate the combination of dabrafenib, a selective BRAF inhibitor, plus trametinib, a selective MEK inhibitor, in a subset of patients with *BRAF* V600–mutant metastatic CRC [143]. This was the first study in patients with metastatic CRC to report prospective testing in PDX models generated from during-study biopsies and correlate the findings with clinical response and, indeed, the response of the PDX models to dabrafenib plus trametinib mimicked the responsiveness of the patient tumor from which they were derived. 

PDX Avatar models can also be studied to understand the mechanisms of tumor resistance to specific drugs and develop predictive biomarkers for those mechanisms. Kopetz et al. reported a phase II pilot study of vemurafenib, an oral BRAF inhibitor, in patients with metastatic *BRAF*-mutated CRC [144]. Concurrently with the trial, they treated a vemurafenib-sensitive *BRAF* V600E PDX model with vemurafenib until disease progression and explored the mechanisms of resistance. They detected co-occurring *KRAS* and *NRAS* mutations at low allele frequency in a subset of the patient tumors (median 0.21% allele frequency) and showed that these mutations were an apparent mechanism of acquired resistance in vemurafenib-sensitive PDX models. These results highlight the robust platform for synchronous co-clinical trials, not only to predict the results of ongoing clinical trials but also to generate clinically relevant rationales to design successful clinical trials. 

The utility of Avatar co-clinical trials is still being established in CRC, and recent studies in other tumor types, such as lung cancer and melanoma, support the concept [145,146,147,148,149,150,151]. Clinical trials to evaluate the utility of PDX Avatar models are ongoing in CRC (NCT02732860 and NCT03263663), sarcomas (NCT02720796), head and neck cancer (NCT02572778), triple-negative breast cancer (NCT02247037), metastatic NSCLC (NCT03134456), pancreatic cancer (NCT02795650), and ovarian cancer (NCT02312245) (Table 1). These studies highlight the potential impact of leveraging annotated PDX models matched with the patients in a clinical trial, and this approach could accelerate development of new drugs and identification of predictive biomarkers.

Although the PDX-patient co-clinical trial approach is highly informative, this platform is time-consuming, labor intensive, and expensive. For instance, patients with advanced cancer are not able to wait the 4 to 8 months required to generate the PDX cohort needed for in vivo experiments. In addition, engraftment success rates are still variable. An alternative approach is to leverage fully annotated 2D or 3D culture models. In CRC, large collections of primary cell lines, organoids, or induced-pluripotent stem cells generated from primary patient tumors have been established and characterized, and can be expanded with ever-greater speed and ease [41,152,153,154,155]. Organoids have the capacity to generate ex vivo distinct organ-like structures that retain the genetic and functional features of CRC and represent these tumors’ diverse heterogeneity. Indeed, CRC organoids are amenable to high-throughput drug screens and have been shown to predict in vivo drug response [152,153,156,157]. This platform could save resources and may complement cell line- and xenograft-based drug studies to help fill the gap between cancer genetics and patient clinical trials.

## 7. Humanized PDX Models Link MSI and Immunotherapy Response in CRC

In CRC, tumors with MSI or hypermutation are characterized by defects in the DNA mismatch repair pathway and classified mainly as CMS1. It has been shown that tumors with MSI and hypermutation display high degrees of immune infiltration with cytotoxic T lymphocytes, activated helper T cells, and natural killer cells [158,159], and these subsets of CRC selectively have high expression levels of multiple immune checkpoints, including PD-1, PD-L1, CTLA-4, LAG-3, and IDO [160,161]. Preliminary studies demonstrated that human immune cells and PD-1-expressing T cells are present in humanized PDX models of CRC [162,163]. Furthermore, in vivo MSI-high and microsatellite-stable (MSS) humanized CRC PDXs have been successfully established [164], and, as expected, characterization of these models confirmed higher levels of immune infiltrates and improved response to the PD-1 inhibitor, nivolumab, in MSI-high PDX models compared to MSS PDX models. These results and others suggest that the MSI subset of CRC may be exceptional responders to checkpoint immunotherapy, and several preclinical and clinical studies are underway to evaluate this hypothesis in CRC [165,166,167,168]. 

Among emerging methodologies to generate humanized mice for immunotherapy studies [169], autologous cell transplantation (ACT), where tumor cells and tumor-infiltrating T cells from the same patient are transplanted sequentially, is widely regarded as holding the most promise. A recent study by Jespersen and colleagues demonstrated that melanoma PDX tumors are eradicated in models where the autologous tumor cells and T cells were derived from a patient who exhibited an objective clinical response to ACT. However, T cells from patients who did not respond to ACT did not kill tumor cells. Taken together, these findings provide a potential framework to develop diverse models of human cancers to assess the efficacy of immunotherapies as well as combination therapies [92].

## 8. Cost–Benefit Analysis of PDX Platform and Alternative Approaches

Although PDX models can better predict drug responses versus traditional cancer cells or cell line-based xenografts, their broad application is limited based on cost and time- and resource-intensiveness. However, recent efforts to use PDXs for in vivo therapeutic screenings of targeted therapies using a single-mouse approach [134] have highlighted the utility of this platform to: (i) identify the best treatment or combination of treatments among a panel of PDXs; and (ii) validate drug effects in prioritized tumor models. Additionally, in “Avatar-guided therapies,” the requirement for PDX expansion is less than for drug screening purposes due to the limited number of approved therapies that can be simultaneously considered for the patient and tested.

Ideal models would possess the cost and scalability of cancer cell lines while preserving the predictive response capacity of PDXs. Patient-derived organoids (PDOs), which are 3D-cultured multicellular aggregates, have emerged in the last few years as an intriguing opportunity to move in this direction [170]. PDOs are derived from pluripotent stem cells or isolated organ progenitors and possess self-renewal and self-organization capabilities and retain the characteristics of the physiological structure and function of their source tissue [152]. In a recent study, Vlachogiannis and colleagues developed PDOs from 110 metastatic tumor samples from 71 patients with colorectal or gastroesophageal cancer enrolled in phase I/II clinical trials [171]. Phenotypic and genotypic profiling of PDOs confirmed similar signatures compared to the original tumors, as well as the same gene-mutation spectrum [172,173]. In addition, molecular profiling of the tumor organoids was matched to drug-screening results, and PDOs predicted tumor response with 88% accuracy and, even more strikingly, predicted failure of a tumor to respond to the drug in 100% of cases evaluated [174]. 

PDOs have great potential for disease modeling for cancer research, and living biobanks of different types of PDOs can be established. Indeed, a robust supply of PDOs derived from cancer patients can be generated quickly and cheaply and, increasingly, researchers are choosing organoids instead of 2D culture models. In short-term conditions, slices of tumor tissue can maintain many in vivo properties, including three-dimensional growth, maintenance of tissue organization/structure, and tumor–immune cells interactions [175]. Further, in the context of innovative patient-specific ex vivo systems, PDOs offer experimental conditions for immunological studies free of murine cells, while maintaining comparable architecture and complexity relative to the human tumor microenvironment. 

One compelling approach is to conduct studies combining tumor organoids and healthy organoids from the same patient to understand the therapeutic index for any particular drug regimen. However, PDOs are not a reliable system to assess therapeutic responses with regard to drug bio-distribution. In addition, there is difficulty in interpreting the medium- to long-term effects of drugs in PDO systems, limiting their predictive capacity for certain drugs with delayed mechanisms of action, such as epigenetic-targeting drugs. Similarly, tumor relapse and drug resistance mechanisms are difficult to model with current PDO technology. Overall, PDOs constitute a valid approach to reduce certain costs/efforts associated to PDX platform and should be always considered as a valuable orthogonal strategy.

## 9. International PDX Biobanking—Opportunity for Multicenter Preclinical Trials

Ideal animal models for preclinical study in oncology should fulfill several criteria to guide clinical decisions: availability in sufficient numbers and variation to fully reflect epidemiological diversity, maintenance of the characteristics of human tumors at histopathologic and molecular levels, and faithful recapitulation of the human therapy response in terms of mechanism of action [10]. To achieve these goals, an increasing number of international groups from industry and academia are working together to develop large, well-annotated PDX cohorts and share data on these models. Annotations should include patient clinical information, histopathologic data, deep molecular profiling, proteomic and metabolomic features, and pharmacologic response data from patients and corresponding PDX models. These systematic efforts have generally been initiated by international multi-institutional consortia, including EurOPDX (Europe), the NCI repository of patient-derived models (US), the Innovative MODels Initiative (IMODI) consortium (France), the Pediatric Preclinical Testing Consortium (US), the Children's Oncology Group cell culture and xenograft repository, the Public Repository of Xenografts (PRoXe) [141], Novartis Institutes for Biomedical Research PDX Encyclopedia (NIBR PDXE) [10,134], and The Jackson Laboratory PDX Resource. These powerful resources will provide opportunities to evaluate new drug/drug combination efficacy and uncover molecular mechanisms of therapy response in a context of population-scale studies [176]. 

Standardizing procedures for generating and biobanking PDX models to achieve reliable and reproducible experimental designs are critical to maximize the impact of studies using these publically available PDX resources. In addition, the platforms to acquire and manage large datasets from PDX models should be harmonized across the research community. To address these points, international PDX experts recently presented the PDX models minimal information standard (PDX-MI) for reporting on the generation, quality assurance, and use of PDX models in cancer research [177]. As one example, Schütte et al. [156] developed a large biobank of 106 CRC tumors, 35 organoids, and 59 xenografts, with extensive -omics data comparing donor tumors and derived models. They demonstrated that the CRC models recapitulate many of the genetic and transcriptomic features of the donors, and that there is a link between molecular profiles and drug sensitivity patterns, including a signature outperforming *RAS/RAF* mutations that predicts sensitivity to EGFR inhibitors. This approach can be applied to molecular classification of CRCs, such as the CMS or CRIS subtypes, so that even if establishment of a particular PDX model fails, different PDX models generated from the same subtype can be shared and utilized as surrogate preclinical models. These multi-institutional PDX biobanking efforts and systematic collaborations will enable us to address the limitations of the current PDX models, facilitating their utility as clinically relevant cancer models.

## 10. Conclusions and Final Remarks

Recent molecular subtyping efforts in CRC have improved our ability to properly stratify CRC patients who will benefit from specific drugs. Humanized PDX models of MSI-high CRC hold great promise for evaluation of novel immunotherapies in the context of a functional human immune system. However, there are still critical considerations and challenges that must be addressed when we apply these models in cancer research, including the amount of time and resources currently required for PDX generation, variable engraftment rates, distinct clonal dynamics and heterogeneity, and lack of elements representing the tumor microenvironment, such as stromal and immune components. Moreover, standardization and harmonization of the platforms across the international community to establish PDX models and manage large-scale integration of molecular profiling data are essential to ensure reliable and reproducible data. We envision that, as these limitations are overcome, PDX models will provide the most clinically relevant experimental platforms to accelerate drug development, guide design of future clinical trials, and eventually support ongoing efforts to make personalized medicine a reality.

## Figures and Tables

**Figure 1 cancers-11-01321-f001:**
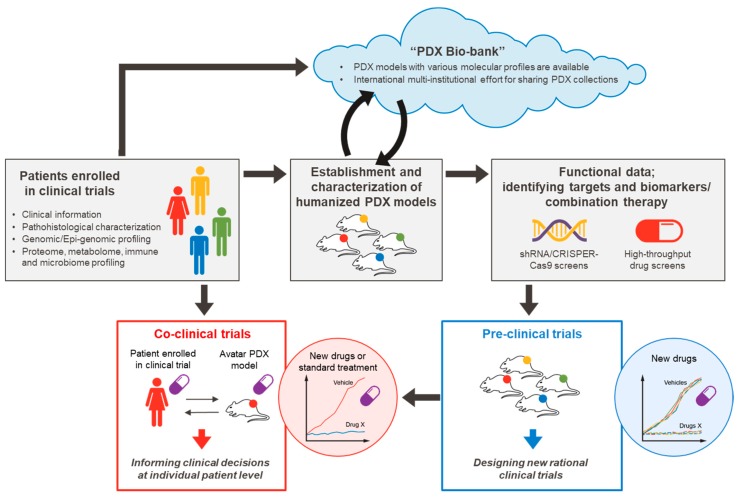
Precision medicine pipeline using patient-derived xenograft (PDX) models.

**Table 1 cancers-11-01321-t001:** Clinical trials involving PDX models.

Identifier	Study Title	Conditions	Status
NCT03358628	Patient-derived Xenograft (PDX) Modeling to Test Drug Response for High-grade Osteosarcoma	Osteosarcoma patients with metastatic relapsed or unresectable progressive disease	New (not yet recruiting)
NCT02247037 ^1^	Patient-derived Xenograft (PDX) Modeling of Treatment Response for Triple Negative Breast Cancer	Women with histologically confirmed triple negative breast cancer	Recruiting
NCT02720796 ^1^	Patient-Derived Xenograft (PDX) Modeling in Adult Patients With Metastatic or Recurrent Sarcoma	Sarcoma	Terminated (recruitment barriers)
NCT02910895	Sarcoma Patient Derived Xenografts (SarcomaPDX)	Soft tissue (non-) metastatic sarcomas	Recruiting
NCT02752893	Estrogen Receptor-Positive Breast Cancer Patient-Derived Xenografts	Breast cancer patients with ER+/HER2− or with a history of ER+/HER2− breast cancer.	Recruiting
NCT02572778 ^1^	Patient-derived Xenograft Models of Tumor From Patients With Head and Neck Cancer (UCL-Xenog)	Squamous cell carcinoma of the head and neck	Recruiting
NCT03134027	Reconstitution of a Human Immune System in a Patient-derived Xenograft (PDX) Model of Genitourinary (GU) Cancers (Immune PDX)	Genitourinary cancer, bladder cancer, kidney cancer, prostate cancer	Recruiting
NCT02732860 ^1^	Personalized Patient-derived Xenograft (pPDX) Modeling to Test Drug Response in Matching Host (REFLECT)	Colorectal neoplasms, colorectal cancer, breast cancer, breast neoplasms, ovarian cancer, ovarian neoplasms	Recruiting
NCT03263663 ^1^	Optimization of Individualized Therapy for CRCs With Secondary Resistance Toward Anti-EGFR Targeted Therapy Using an Avatar Model	Colorectal cancer	Recruiting
NCT03134456 ^1^	Pembrolizumab for Metastatic NSCLC Patients Expressing PD-L1 Who Have Their Own PDX	Metastatic non-small cell lung carcinoma expressing PD-L1 after failure of platinum-based combination chemotherapy.	New (not yet recruiting)
NCT02752932	TumorGraft-guided Therapy for Improved Outcomes in Head and Neck Squamous Cell Cancer: A Feasibility Study	Head and neck squamous cell carcinoma	Terminated (completed)
NCT01750164	Patient-derived Breast Cancer Xenografts	Invasive breast cancer with metastatic disease	Terminated (discontinued funding)
NCT03336931	PRecISion Medicine for Children With Cancer (PRISM)	Childhood solid tumor, childhood brain tumor, childhood leukemia, refractory cancer, relapsed cancer	Recruiting
NCT01130571	Establishment and Characterization of Patient-derived Non–Small Cell Lung Cancer Xenografts and Cell Cultures	Non–small cell lung cancer	Unknown (Verified January 2014)
NCT02795650 ^1^	Personalised Therapy for Patients With Metastatic Adenocarcinoma of the Pancreas Determined by Genetic Testing and Avatar Model Generation (AVATAR)	Pancreatic adenocarcinoma	Recruiting
NCT02312245 ^1^	Avatar-Directed Chemotherapy in Treating Patients With Ovarian, Primary Peritoneal, or Fallopian Tube Cancer	Recurrent fallopian tube carcinoma, recurrent ovarian carcinoma, recurrent primary peritoneal carcinoma	Recruiting

^1^ Avatar PDX-oriented clinical trials.

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
