# Peer review of "Current and Future Horizons of Patient-Derived Xenograft Models in Colorectal Cancer Translational Research"

_cancers, 2019, doi:10.3390/cancers11091321_

Round 1

Reviewer 1 Report

Overall I think this is a well written and extremely comprehensive review of the literature on PDX. In particular the paper strikes a good balance between detail of certain studies whilst covering the breadth of issues/research. The reference base for this narrative review is certainly up to date.

I do think however that much (~50-60%) of the issues highlighted in the paper have been summarised in previous reviews on the topic. Where this paper is more relevant, is in the discussion of the last 5 or so years worth of research which has more comprehensively described molecular characteristics of PDX vs. donor tumours, studies on the tumour microenvironment, humanised models, organoid-PDX models for high throughput, and international efforts to standardise grafting and PDX-trial protocols (via PDX banks etc.). The manuscript would have greater impact by focusing more on these aspects rather than revisiting other issues that have been previously documented i.e. an update since the last significant review.

Furthermore, I think the general tone is probably a bit too optimistic in the potential for PDX as a platform. My recommendation is that this should be offset by having a specific subtitle/subsection related to ongoing limitations of PDX (some of which a alluded to but somewhat lost throughout the body text). On the topic of intratumoural heterogeneity - I think this is only a limitation depending what your reserach question is; if you are interested in mechanisms then the PDX "black box" is a hinderance, if you want to replicate what happens in the "black box" of the patient's tumour eg. in a drug trial, then this is potentially an advantage. I also think there should be more reference the use of organoid-PDX/PDO models as a means of potentially overcoming some limitations of the traditional PDX platform.

Lastly, the paper outlines the issues related to PDX somewhat by describing papers that happen to have used a CRC tumour stream to study, rather than how to the traits of PDX models address issues that pertains to specifically CRC eg. CRC is molecular a very diverse disease (only recently with a consensus molecular classification), with a complex immune/stromal/microbiome microenvironment, and as is illustrated by the work of Calon/Isella and others, the stroma (a compartment lost by the PDX after a few passages) probably plays a significant role in primary CRC ability to invade and thus metastasize, MSI-hi (and thus immuno-oncological studies) are actually a relatively small proportion of human cases with more favorable prognosis...I think inverting the focus from PDX > CRC to CRC > PDX would make this paper much more clinically relevant.

My personal feeling is that that the resources and time required to build up a statistically sound model precludes PDX from ever really being used as useful real-time "Avatars", but what they do offer is, despite their limitations, a means to multiplex treatment groups in a way not possible with human trials (esp. with PDOs).

In summary; a well written and comprehensive paper definitely worth being published.

1. Should have a specific section on limitations of PDX models.

2. Would benefit from spending a greater proportion of the word count on progress since the last review

3. Would benefit from focusing on the issues around PDX as to how they specifically pertain to CRC.

Author Response

We thank the reviewer for appreciating our work, as well as acknowledging the importance of the study from a translational point of view. Moreover, we agree with our reviewer that a significant portion of this manuscript is dedicated to reviewing PDX issues for translational studies, many of which have been raised and covered in previous works. However, our hope and intent is that, by providing a deeper analysis of these issues specifically in the CRC context (which is molecularly and therapeutically distinct from many of the solid tumors extensively modeled through PDX efforts), this review will serve members of the scientific community interested in accelerating basic findings using colorectal cancer PDXs.

A second concern raised by the reviewer is related to the limitations of PDXs. We fully agree with this comment, and we have included a dedicated cost-benefit section in the revised manuscript (see “Cost-benefit analysis of PDX platform and alternative approaches”). In this paragraph, we extensively cover limitations and constrains associated with the PDX platform, and we present patient-derived organoids (PDOs) as a potential alternative strategy to reduce cost, animals and effort. Limits and downsides of PDOs are discussed as well.

In relation to the third point raised by the reviewer regarding shifting the review’s focus to describe how PDX models address issues that pertain to CRC, we believe that our current presentation has tried to do this whenever possible. We often begin by introducing historic efforts around PDX modeling and characterization in CRC, and we have attempted to present data and drive discussions in the context of which current or future directions are needed to achieve clinical relevance for the disease (something that is constantly evolving). Specifically, in section 3, where we introduced CMS subtyping, we clearly discussed the importance of profiling PDX models to CMS4, which is characterized by extensive stromal components. Even if human stromal cells are substituted by mouse counterparts in PDXs, these models provide a unique tool to investigate drug responses and molecular insights related to all the CMS subtypes, something that Isella and colleagues exploited to define features of this mesenchymal transcriptional subtype. In addition, several clinical questions cannot be answered at the moment simply by adopting the PDX platform, but will require more integrated solutions and technological advancements that demand in-depth discussions that would have shifted the focus of this work away from PDXs (which is the focus of this special issue on Tumor Xenografts).

Reviewer 2 Report

This is a fairly comprehensive and contemporary review of the state of play with using PDXs in CRC tarnslational research.

It is on the whole well written and I think with a couple of issues addressed would be a worthwhile addition to the literature on PDXs.

Firstly there is little consideration of the 3Rs in this review. Clearly the poor take rate and numbers of animals required in generating sufficient tumours to use in studies means that this technique has a high cost in terms of animal use. The cost-benefit should be considered in comparison with other in vivo models.

Secondly whilst some promising reports of good correlation between results in PDX and clinical studies is made (especially section 5.1), little is done to put this in context compared to other in vivo models, e.g. the 'standard' cell line xenograft model. I think including this would strengthen the case for PDXs.

Author Response

We would like to thank our reviewer for the accurate and positive evaluation of our work. The points raised are well-taken and have contributed to strongly improve and consolidate the new version of the manuscript. In details:

We have included a dedicated cost-benefit section (See Cost-benefit analysis of PDX platform and alternative approaches) in the revised manuscript. In this paragraph, we extensively cover limitations and constrains associated with the PDX platform, and we present patient-derived organoids (PDOs) as a potential alternative strategy to reduce cost, animals and effort. Limits and downsides of PDOs are discussed as well.

We have also introduced a brief paragraph focused on the ability of cell line-based xenografts to predict clinical outcomes in comparison to PDXs. As suggested by our reviewer, this strengthens the case for PDXs. Differences between the two systems that might justify choosing one platform over the other are reported and highlighted.

Reviewer 3 Report

The review of CRC PDX models by Inoue and colleagues is very convincing and up-to-date. 

I have only one suggestion towards a blind spot concerning the humanized PDX. They promise best results when used in autologous settings - i.e., tumors and lymphocytes from the same patient(s). Mentioning this would be worthwhile.

Author Response

We thank the reviewer for appreciating our work and for the extremely useful suggestion/insight. We have included a short paragraph on autologous humanized PDXs in the appropriate section of the revised review.